# Atomic Simulations of U-Mo under Irradiation: A New Angular Dependent Potential

**Wenhong Ouyang, Wensheng Lai, Jiahao Li, Jianbo Liu \* and Baixin Liu**

The Key Laboratory of Advanced Materials (MOE), School of Materials Science and Engineering, Tsinghua University, Beijing 100084, China; oywh18@mails.tsinghua.edu.cn (W.O.); wslai@mail.tsinghua.edu.cn (W.L.); lijiahao@mail.tsinghua.edu.cn (J.L.); dmslbx@mail.tsinghua.edu.cn (B.L.)
\* Correspondence: jbliu@mail.tsinghua.edu.cn

**Abstract:** Uranium-Molybdenum alloy has been a promising option in the production of metallic nuclear fuels, where the introduction of Molybdenum enhances mechanical properties, corrosion resistance, and dimensional stability of fuel components. Meanwhile, few potential options for molecular dynamics simulations of U and its alloys have been reported due to the difficulty in the description of the directional effects within atomic interactions, mainly induced by itinerant f-electron behaviors. In the present study, a new angular dependent potential formalism proposed by the author's group has been further applied to the description of the U-Mo systems, which has achieved a moderately well reproduction of macroscopic properties such as lattice constants and elastic constants of reference phases. Moreover, the potential has been further improved to more accurately describe the threshold displacement energy surface at intermediate and short atomic distances. Simulations of primary radiation damage in solid solutions of the U-Mo system have also been carried out and an uplift in the residual defect population has been observed when the Mo content decreases to around 5 wt.%, which corroborates the negative role of local Mo depletion in mitigation of irradiation damage and consequent swelling behavior.

**Keywords:** uranium; molybdenum; molecular dynamics; interatomic potential

## 1. Introduction

Pure uranium (U) is limited to a considerable extent in the production of fuel elements due to its questionable properties. $\alpha$-U, which is stable at low temperatures, has pronounced anisotropy and comparatively low strength characteristics at elevated temperatures. Meanwhile, the existence of high-temperature body-centered cubic (bcc) $\gamma$-U and intermediate tetragonal $\beta$-U indicates harmful allotropic transformations during cyclic heat treatment. Moreover, U shows high chemical activity in corrosion behaviors [1,2]. A common solution of problems above is the utilization of uranium alloys, which retains the isotropic structure of $\gamma$-U at room temperature. So far, molybdenum (Mo) and zirconium (Zr), along with a few other elements appear to be promising options in U alloys for nuclear fuel. In particular, alloying U with Mo remarkably improves its mechanical properties, corrosion resistance, and dimensional stability, along with providing other benefits such as high thermal conductivity and low thermal expansion [3,4]. Enhanced mechanical properties enable U-Mo alloys to be utilized for the production of cores of fuel elements of arbitrary configurations. High corrosion resistance in water of high parameters and reliable dimensional stability under irradiation are deciding factors governing the choice of U-Mo alloys as fuel materials. It is also deemed expedient in all cases for U-Mo alloys to be included in the construction of fast-neutron reactors [1].

However, the introduction of alloy elements also brings about some peculiar physical phenomena whose fundamental mechanism remains to be understood. During the cooling of U-Mo alloys, several metastable phases have been observed in a sequence of bcc $\gamma \rightarrow$ bcc with doubled lattice constant $\gamma^S \rightarrow$ body-centered tetragonal (bct) $\gamma^0 \rightarrow$ monoclinic

$\alpha'' \to$ orthorhombic $\alpha'$ [5,6]. The addition of Mo was suggested to stiffen the U lattice against shear, thus hindering and also complicating the transition progress. However, during service life, Mo has been observed to be depleted from grain boundaries (GBs) among a wide range of compositional banding [7], which could lead to an onset of phase decomposition. In particular, the elemental redistribution near GBs could also be implicated in the generation of irradiation-induced recrystallization (IIR) [8], where fuel grains are subdivided into nano-sized grains from the GBs during service life. Meanwhile, IIR is suggested as an important culprit behind accelerated swelling behavior of nuclear fuel alloys [9], which enhances the reach of GBs into the fuel grains, destroying low swelling intra-granular fission gas bubbles (FGBs) and producing high swelling inter-granular FGBs.

To shed light on the underlying mechanism of microstructure evolution under irradiation at U-Mo fuel operation temperatures, constructing corresponding potential for molecular dynamics simulation has been under research in recent years. An embedded atom method (EAM) potential for U-Mo-Xe has been developed by the force matching method [10], through which the properties of U2Mo and $\alpha$-U-Mo could be reproduced well, even without taking them into account in the fitting process. However, traces of overfitting were observed, suggesting the limitation of EAM for the description of a U-Mo system. By the use of the interatomic potential proposed by Smirnova et al. [10], attempts to reveal cascade effects on residual defects in U-Mo alloys have also been reported [11]. Also based on the U-Mo-Xe EAM potential, Hu et al. [12] have investigated the relationship between the pressure, equilibrium Xe concentration, and radius of Xe bubbles in U-10 wt% Mo by molecular dynamics (MD) simulations. Utilizing a formalism improved on the basis of EAM, namely Angular-Dependent Potential (ADP), Smirnova et al. have qualitatively reproduced the properties of cubic and tetragonal phases of $\gamma$-U-Mo alloys [13] and improvements were also made in reproduction of the density, coefficient of thermal expansion, and diffusion behavior [14]. It was also suggested that a successful capture of atomic properties in $\gamma$-U and $\gamma$-U-Mo systems requires utilization of a potential form, with its level of complexity no lower than the ADP. Moreover, Starikov et al. [15] have also constructed an ADP with the same form as that developed by Smirnova, which paid special attention to the description of metastable phases of U-Mo solid solution in the fitting process.

For the present work, we first constructed a U-Mo interatomic potential using the ADP formalism proposed by the author's group [16]. The ADP formulas were further modified such that it can more accurately reproduce the threshold displacement energy surface as well as many-body repulsion at intermediate and short interatomic distances. We then applied the obtained potential in MD simulations to study the U-Mo system under irradiation.

## 2. Materials and Methods

Based on the formalism of EAM, several modified versions including the MEAM and ADP potentials were developed by the addition of angular dependent terms to increase the description accuracy. Similarly, for the present study, the new angular-dependent interatomic potential calculated atomic energy contribution $E_i$ by the following formula:

$$E_i = \frac{1}{2} \sum_{j \neq i} \phi_{ij}(r_{ij}) + F(\overline{\rho}_i) - \vartheta_i \tag{1}$$

where

$$\begin{aligned} \phi_{ij}(r_{ij}) &= \sum_{i=0}^{3} a_i (r_{\mathrm{c}} - r_{ij})^{3+i}, \\ F(\overline{\rho}_i) &= -\sqrt{\overline{\rho}_i}, \overline{\rho}_i = \sum_{j \neq i} \rho_{ij}(r_{ij}), \\ \rho_{ij}(r_{ij}) &= \alpha^2 (r_{\mathrm{c}} - r_{ij})^4 \exp\left[-\beta\left(\frac{r_{ij}}{r_0} - 1\right)^2\right], \vartheta_i = \vartheta_i^{\mathrm{u}} + \vartheta_i^{\mathrm{y}} + \vartheta_i^{\mathrm{w}}. \end{aligned} \tag{2}$$

It could be indicated by the first two terms in Equation (1) that the main part of the atomic energy maintains the form of the widely used EAM potential. To be exact, $\phi_{ij}$ and

$F(\overline{\rho}_i)$ describe the atomic interactions of the electrostatic repulsion between ion cores and the attraction induced by embedding atoms in the electron atmosphere provided by their neighbor atoms, respectively. In calculation of $E_i$ of the *i*-th atom, summation is over its *j*-th neighbor atom within the cutoff distance $r_c$, which was set to 4.7 Å in the present study. In particular, $r_0$ and $r_{ij}$ denotes the first neighbor distance in a reference lattice at equilibrium and the distance between the *i*-th center atom and its *j*-th neighbor atom, respectively. Due to the inherent sphere symmetry in the potential form, EAM poorly describes some peculiar atomic behaviors in systems of U alloys, such as the stability of the alpha phase with low symmetry at room temperature. In the present work, the description of the atomic energy was modified with the addition of a new angular-dependent term $\vartheta_i$, which has three components given as the following:

$$\vartheta_i^{u} = \sum_{\alpha} \left( \sum_{j \neq i} \psi_j^{u}(r_{ij}) \frac{r_{ij}^{\alpha}}{r_{ij}} \right)^2,$$

$$\vartheta_i^{v} = \sum_{\alpha,\beta} \left( \sum_{j \neq i} \psi_j^{v}(r_{ij}) \frac{r_{ij}^{\alpha} r_{ij}^{\beta}}{r_{ij} r_{ij}} \right)^2 - \frac{1}{3} \left( \sum_{j \neq i} \psi_j^{v}(r_{ij}) \right)^2,$$

$$\vartheta_i^{w} = \sum_{\alpha,\beta,\gamma} \left( \sum_{j \neq i} \psi_j^{w}(r_{ij}) \frac{r_{ij}^{\alpha} r_{ij}^{\beta} r_{ij}^{\gamma}}{r_{ij} r_{ij} r_{ij}} \right)^2 - \frac{3}{5} \sum_{\alpha} \left( \sum_{j \neq i} \psi_j^{w}(r_{ij}) \frac{r_{ij}^{\alpha}}{r_{ij}} \right)^2 \tag{3}$$

where

$$\psi_j^{t}(r) = c_t exp\left( -\frac{(r - d_t)^2}{2\lambda_t^2} \right), t = u, v, w. \tag{4}$$

In Equation (3), superscripts $\alpha, \beta, \gamma = 1, 2, 3$ refer to the Cartesian components of position vectors. $\vartheta_i^{u}$, $\vartheta_i^{v}$, and $\vartheta_i^{w}$ denote the angular-dependent components of the first, second, and third order, with $\psi_j^{t}$ as basis function in the form of the normal distribution function. In the present study, it was assumed that all the three angular-dependent components are required in the fitting process for the description of U and the first two ones for Mo.

A total of 15 parameters were adjusted in the fitting process, which include $a_i, \alpha, \beta, c_t, d_t, \lambda_t$ listed in Table 1. In particular, the angular-dependent parameters of the third order for Mo were kept to zero. During the fitting, an error function was optimized. Macroscopic properties including the cohesive energy, lattice parameters, and elastic constants of certain lattices were calculated with a set of fitting parameters and compared with reference data from the literature or calculations by the First Principle (FP) method. The framework of the fitting procedure was implemented in Python, where symbolic computation and optimization are supported by Sympy and Scipy packages. In particular, in order to improve the description of point defects, several lattices with certain point defects were included in the reference data and allowed to relax during the calculations of the formation energies.

In order to capture atomic behaviors within equilibrium distances, $\phi_{ij}$ and $\rho_{ij}(r_{ij})$ within the inner cutoff were further modified. Interpolations are implemented for $\rho_{ij}(r_{ij})$ to approach a constant and for $\phi_{ij}$ to connect smoothly with the well-known Ziegler-Biersack-Littmark (ZBL) potential [17]. In particular, special attention was paid to the intermediate repulsive range in development of the potential for further employment in cascade simulations. Traditionally speaking, a smooth interpolation is used between the pairwise part of many-body potentials in the near-equilibrium range and the ZBL potential in the short range. Moreover, in 2016, Stoller et al. [18] pointed out that no accepted standard method had been developed in the interpolation process and that how the force fields are linked can sensibly influence the results of cascade simulation. Similar to the modifications implemented by Stoller, the present study also corrected the potential with calculation results from density functional theory (DFT) as a benchmark. The DFT calculations were performed using the Vienna ab initio simulation package (VASP) [19] with projector augmented wave pseudo-potentials. A kinetic energy cutoff of 400 eV and a $4 \times 4 \times 4$ grid with the Monkhorst−Pack method were employed. The ADP calculations were performed using the Large-scale Atomic/Molecular Massively Parallel Simulator (LAMMPS) [20]. A supercell containing $3 \times 3 \times 3$ conventional bcc cells under equilibrium

($a_U$ = 3.48 Å) was chosen for calculations of energetic benchmark and then an atom in the supercell was translated in several equidistant steps toward its neighbors along the direction of <110> and <100>.

**Table 1.** Fitted parameters of ADP for U–Mo system.

| Parameter | U–U | Mo–Mo | U–Mo |
|---|---|---|---|
| $a_0$ (eV·Å$^{-3}$) | −1.41881 | 1.75876 | 6.095612 |
| $a_1$ (eV·Å$^{-4}$) | 2.499597 | −3.74227 | −11.667 |
| $a_2$ (eV·Å$^{-5}$) | −1.40956 | 2.276131 | 7.291495 |
| $a_3$ (eV·Å$^{-6}$) | 0.279328 | −0.42748 | −1.48751 |
| $\alpha$ (eV·Å$^{-2}$) | 0.86737 | 0.255577 | - |
| $\beta$ | −0.88394 | 2.017747 | - |
| $c_u$ (eV$^{1/2}$) | 0.083778 | 0.369529 | −0.2 |
| $d_u$ (Å) | 2.825854 | 2.590854 | 3.173449 |
| $\lambda_u$ (Å) | 0.151763 | 0.62249 | 0.312741 |
| $c_v$ (eV$^{1/2}$) | 0.015177 | 0.225399 | 0.2 |
| $d_v$ (Å) | 3.153919 | 2.791784 | 3.208256 |
| $\lambda_v$ (Å) | 0.149995 | 0.186653 | 0.409074 |
| $c_w$ (eV$^{1/2}$) | 0.133803 | - | −0.2 |
| $d_w$ (Å) | 3.402176 | - | 2.952746 |
| $\lambda_w$ (Å) | 0.359213 | - | 0.167571 |
| $r_c$ (Å) | 4.7 | 4.7 | 4.7 |
| $r_0$ (Å) | 2.7408 | 2.7387 | - |

For simplicity in description of energetic variance, the present study used $\Delta E_t$ as the following:

$$\Delta E_t = E_{\text{translated}} - E_{\text{perfect}} \qquad (5)$$

where $E_{\text{translated}}$ is the energy of the configuration with atomic translation and $E_{\text{perfect}}$ is the energy of the original configuration with perfect symmetry. The energetics of the supercell as a function of the ratio between the distance of the nearing dimers and the lattice constant are shown in Figure 1. It could be seen that a reasonable reproduction of energetics variance was obtained in the intermediate range, especially for U-U and U-Mo dimers. Moreover, it should be noted that in cases of dimers with different elemental components, the choice of which atom is moved toward the other produces similar but actually different energetics variances. The difference depends mainly on the distinct interactions between the translated atom and its atomic environment. Good reproduction was achieved in the present study, indicating reasonable accuracy of the refitted potential in the intermediate range. Meanwhile, sensible deviations could be seen at the outer end of the intermediate range, especially for Mo-Mo dimers, which are induced by the difficulty of resolving the energetic uplift at the outer end of the intermediate range while leaving the description by the ADP of near-equilibrium atomic interactions intact. Nonetheless, even under the deviations, the general curve shapes of energetics variance were reserved to a large extent, indicating a good description of dynamic properties during atomic interactions in the intermediate range.

The newly developed potential was then applied in MD simulations of displacement cascades in U-Mo alloys in the present study. Simulation boxes containing $60 \times 60 \times 60$ bcc unit cells were set up under periodic boundary conditions (PBC) with up to 648,000 atoms included. Alloying elements were randomly distributed in initial lattices before simulations. A total of four proportions of alloying elements, that is, 0 wt.%, 2.5 wt.%, 5 wt.%, 7.5 wt.%, and 10 wt.%, were chosen to evaluate the effect of the content of the alloying element on displacement cascades. Given the high symmetry of the bcc lattice, a total of 15 evenly distributed directions were sampled in the fundamental orientation zone, which is shown in Figure 2. The initial kinetic energies of the primary knock-on atoms (PKA) was set to 5 keV for all the simulations. For each set of initial conditions, simulations were carried out up to eight times to reduce statistical error in the following analysis.

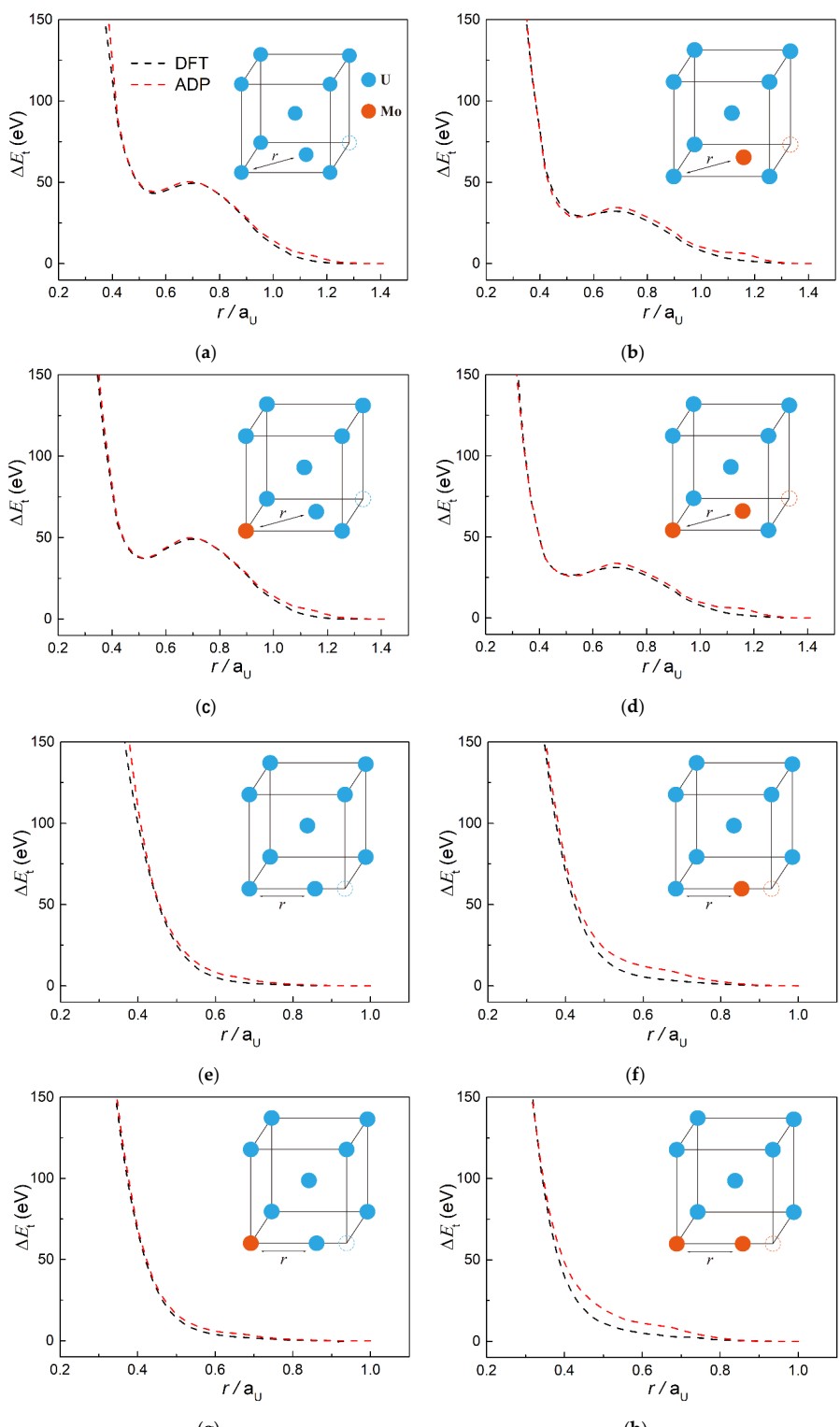

**Figure 1.** The energetics of the supercell as a function of the ratio between the distance of the nearing dimers and the lattice constant of bcc U. In particular, nearing atom pairs include (**a**) U–U in <110>, (**b**) U–Mo in <110>, (**c**) Mo–U in <110>, (**d**) Mo–Mo in <110>, (**e**) U–U in <100>, (**f**) U–Mo in <100>, (**g**) Mo–U in <100> and (**h**) Mo–Mo in <100>. The black and red dash lines represent calculation results from DFT and MD using ADP. In the configuration diagrams, blue and orange circles represent U and Mo atoms, respectively, and the original positions of displaced atoms are indicated by dotted circles.

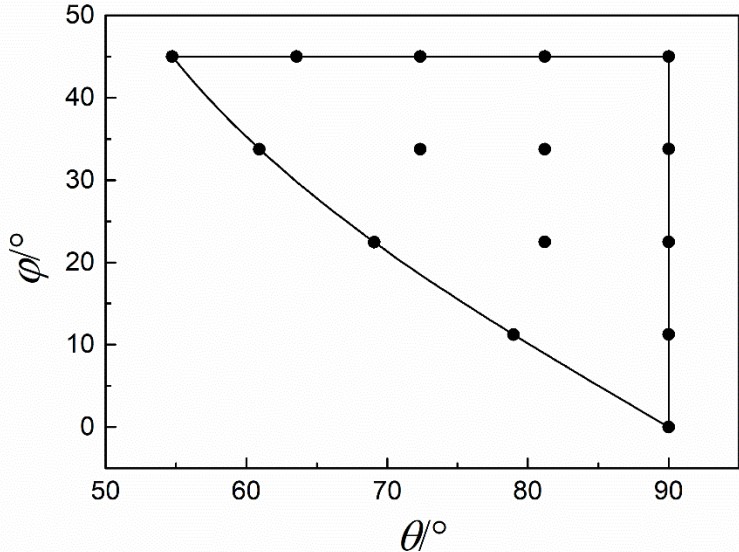

**Figure 2.** Angular distribution of launch directions sampled in the fundamental orientation zone of bcc lattice.

Prior to each set of cascade simulations, the simulation block was equilibrated under 600 K and 0 GPa in an isothermal-isobaric (NPT) ensemble until the temperature and pressure reached a stable level. The initial structures for cascade simulations were chosen randomly from those under equilibrium in a series of time steps. During the cascade simulations, the simulation block was divided into three layers. In the outmost layer, the atoms within 2 Å from the boundary were maintained as static all along the simulation to avoid excessive overall displacement of the simulation block. Atoms located at a distance of 2 to 8 Å from the boundary were partitioned into the second layer. In this layer, a thermostat was applied by rescaling the velocities of the atoms to maintain a stable level of temperature and to absorb the excess kinetic energy from the cascade. The rest of atoms in the core region were simulated with the constant NVE ensemble and a variable time step in a range of 0.01 to 1 fs was used in all simulations, which depends on the maximum of atom displacements between consecutive steps.

To determine and record defect sites during simulations, some native commands in LAMMPS were invoked, which construct a voronoi tessellation dividing each atom into an exclusive cell at the start of each simulation. In particular, at equal intervals of timesteps, voronoi cells with occupancy not equal to 1 were recorded with their position coordinates, while those with occupancy equal to 0 and larger than 1 were marked as vacancy and interstitial sites, respectively. In cluster analysis, adjacent defect sites within a cutoff radius of 4.1 Å are classified into the same cluster group and the size of a cluster is defined as the total count of defect sites within the cluster. By using other commands in LAMMPS, atom displacements and element types were also recorded. The program OVITO [21] was utilized for data extraction and statistical analysis.

## 3. Results

### 3.1. Fitting Results

In the fitting procedure, reference data were gathered from experimental results in the literature and FP calculations as supplements, especially for phases not observed in experiments. Fitted parameters are shown in Table 1. The functions defining the newly developed ADP are plotted in Figure 3.

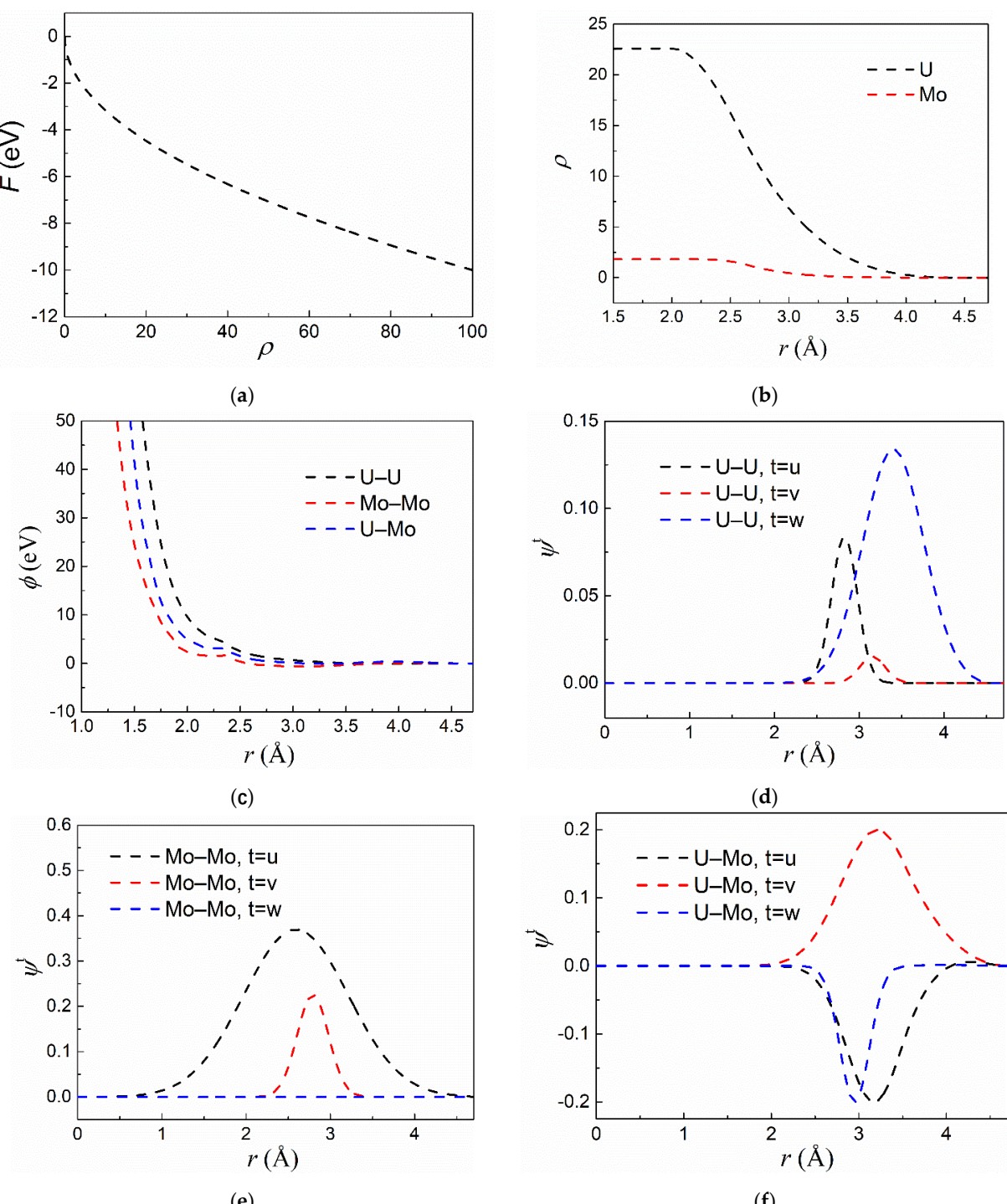

**Figure 3.** Potential functions of the newly developed ADP. Embedding function $F$, electron density function $\rho$ and pair potential $\phi$ are shown in (**a**–**c**), respectively; angular-dependent terms of $\psi^u$, $\psi^v$ and $\psi^w$ are grouped together and plotted for elemental pairs of (**d**) U–U, (**e**) Mo–Mo and (**f**) U–Mo.

Fitting results of pure phases of U and Mo are shown in Tables 2 and 3. It could be seen that the lattice parameters were reproduced moderately well and the hierarchy of relative stability of allotropes was correctly reflected in terms of the cohesive energies. During fitting, specific attention was paid to the reproduction of elastic constants and the formation energies of defects. The fitting results of cross potentials of U–Mo are given in Table 4. During the fitting of the U–Mo cross potential, two configurations were

added to the database, since only the tetragonal phase of $U_2Mo$ has previously been found in experiments.

**Table 2.** Reproduced values of lattice constants, cohesive energies, and elastic constants of pure uranium.

| Structure | Properties | Present Work | MEAM [22] | ADP [15] | Experiment | FP |
|---|---|---|---|---|---|---|
| α-U | $E_{coh}$ (eV/at) | 5.46 | 5.547 | 4.23 | 5.550 [23] | - |
| | a (Å) | 2.881 | 2.721 | 2.849 | 2.836 [24] | - |
| | b (Å) | 5.486 | 6.381 | 5.841 | 5.867 [24] | - |
| | c (Å) | 5.221 | 4.858 | 4.993 | 4.935 [24] | - |
| | y | 0.108 | 0.093 | 0.103 | 0.102 [24] | - |
| | B (GPa) | 153 | 143 | 147 | 136 [25] | - |
| | $E_v$ (eV) | 1.17 | 2.597 | - | - | 1.95 [26] |
| γ-U | $\Delta E_{bcc \to ort}$ (eV/at) | 0.01 | 0.15 | 0.09 | - | 0.278 |
| | a (Å) | 3.479 | 3.463 | 3.52 | 3.47 [27] | 3.455 |
| | $C_{11}$ (GPa) | 128.2 | 144.0 | 183.6 | - | 103.0 |
| | $C_{12}$ (GPa) | 124.1 | 49.0 | 92.8 | - | 142.0 |
| | $C_{44}$ (GPa) | 38.2 | −36.2 | 79.9 | - | 46.0 |
| | $E_v$ (eV) | 1.34 | - | - | - | 1.38 |
| | $E_i$ (eV) | 1.08 | - | - | - | 0.9 |

**Table 3.** Reproduced values of lattice constants, cohesive energies, and elastic constants of pure molybdenum.

| Structure | Properties | Present Work | FP | Experiment [28–30] |
|---|---|---|---|---|
| bcc-Mo | $E_{coh}$ (eV/atom) | 6.349 | 6.290 | - |
| | a (Å) | 3.171 | 3.162 | 3.147 |
| | $C_{11}$ (GPa) | 473.9 | 488.8 | 465 |
| | $C_{12}$ (GPa) | 143.3 | 146.6 | 176 |
| | $C_{44}$ (GPa) | 67.0 | 108.3 | - |
| | $E_v$ | 2.72 | 2.723 | 2.6–3.2 |
| fcc-Mo | $\Delta E_{bcc \to fcc}$ (eV/atom) | 0.439 | 0.327 | - |
| | a (Å) | 4.156 | 4.004 | - |
| hcp-Mo | $\Delta E_{bcc \to hcp}$ (eV/atom) | 0.628 | 0.328 | - |
| | a (Å) | 2.842 | 2.948 | - |
| | c (Å) | 4.212 | 4.263 | - |

**Table 4.** Reproduced values of lattice constants, cohesive energies, and elastic constants of several U–Mo binary phases.

| Structure | Properties | Present Work | FP |
|---|---|---|---|
| tetra-$U_2Mo$ | $E_{coh}$ (eV/at) | 6.044 | 6.361 |
| | a (Å) | 3.304 | 3.427 |
| | c (Å) | 9.921 | 9.833 |
| | y | 0.329 | 0.328 |
| bcc-$U_{15}Mo$ | $\Delta E_{tetra \to bcc}$ (eV/at) | 0.391 | 0.116 |
| | a (Å) | 6.861 | 6.84 |
| | $C_{11}$ (GPa) | 159.3 | 124.4 |
| | $C_{12}$ (GPa) | 134.7 | 140.6 |
| P6-$U_2Mo$ | $\Delta E_{tetra \to P6}$ (eV/at) | 48.4 | 34.9 |
| | a (Å) | −0.119 | −0.035 |
| | c (Å) | 4.823 | 4.818 |

### 3.2. Cascade Simulations

Typical time evolutions of defects in U-Mo alloy under different Mo contents are given in Figure 4. Similar curves were observed in all cases, in which the counts of defects increase rapidly to a peak value and decrease in a gradually reducing speed. Overall, the addition of Mo increases residual defect populations and Figure 5 shows that with the Mo content, increasing high percentages of residual defects were distributed in large clusters. It could then be inferred that Mo plays a negative role in the recrystallization process of the cascade damage, which in the case of pure U lattice only produces dispersed residual defects in a low population. Considering the inherent impurity of Mo, the recrystallization of bcc lattice with high symmetry might have been interfered with by the local stress and distortion provided by solute Mo atoms.

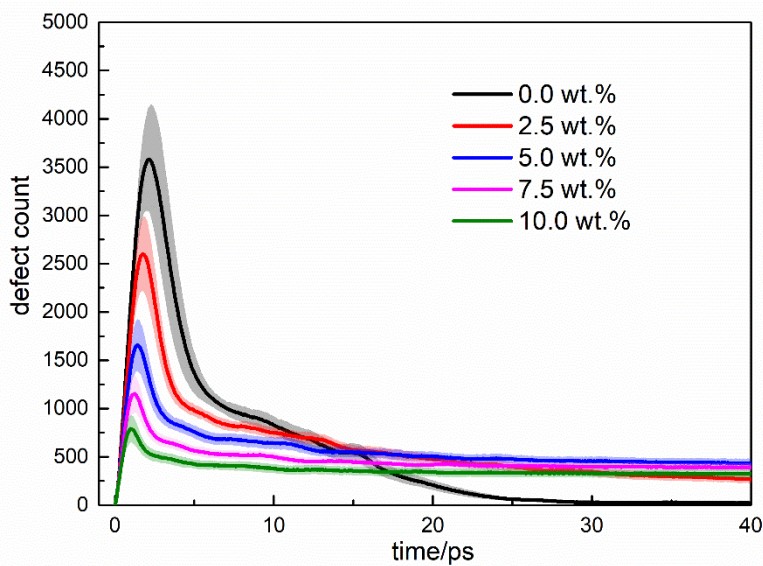

**Figure 4.** Time evolutions of the defect count in U-Mo systems with different Mo contents.

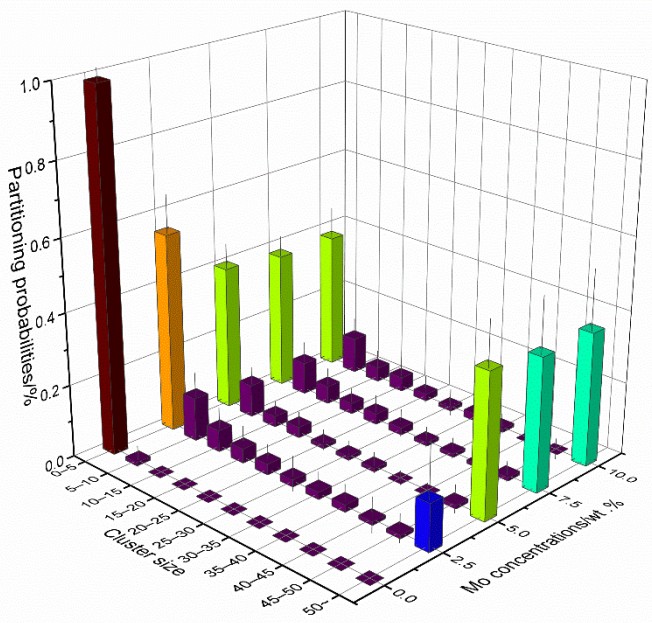

**Figure 5.** Size distributions of residual defect clusters versus Mo contents.

However, as shown in Figure 6a, a remarkable fall in residual defect populations was found when Mo contents exceeded 5 wt.%, suggesting that the decrease of Mo content

from that in fuel alloys (>7 wt.%) brings about more irradiation damage with a potential acceleration of defect evolution. Meanwhile, the onset of IIR was suggested to be highly correlated with the accumulation of defect structures near GBs [31]. It could be further inferred that the local depletion of Mo in U-Mo potentially has an important influence on IIR and even accelerates swelling behavior from the primary stage. Actually, in U-Mo alloys with a relatively low Mo content (U-7 wt.% Mo), an increased swelling has already been observed [32].

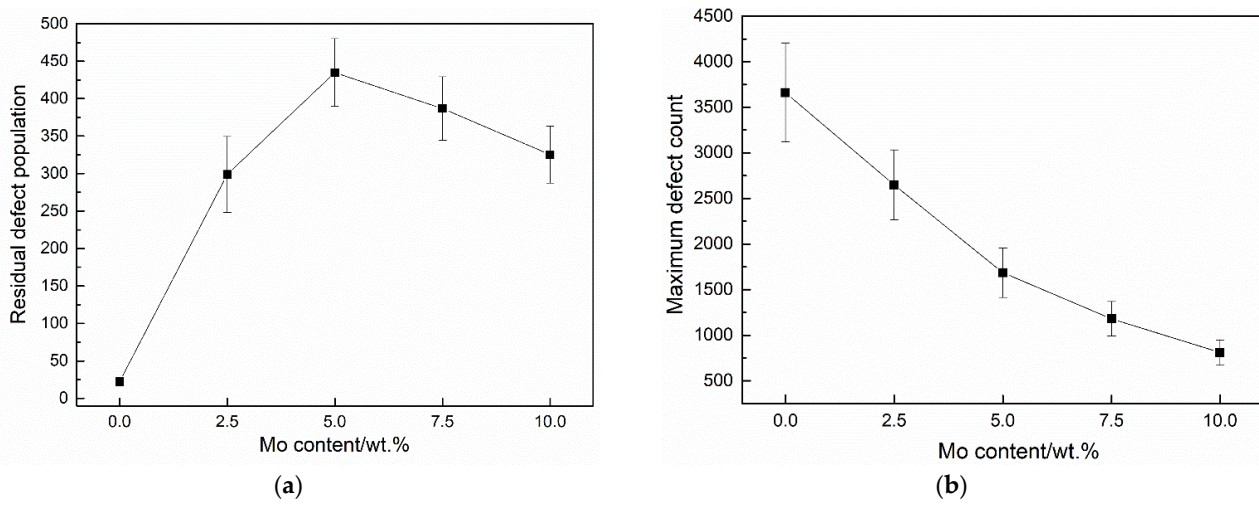

**Figure 6.** Statistical results of (**a**) residual defect population and (**b**) peak values of defect population during cascade simulations with different Mo contents.

## 4. Discussion

To investigate the underlying mechanism of the reduction of residual defect population, further attention was first paid to the spatial distribution of residual defects. Figure 7a shows the average distances between residual defects and the initial position of PKA as a function of Mo content and it could clearly be seen that the increase of Mo content restricts the spatial distribution of residual defects. Meanwhile, it could also be noted based on Figure 6b that the peak values of defect population decrease as Mo contents increase, suggesting that cascade processes are considerably limited in smaller volumes with a higher mass fraction of Mo. In general, a lattice with a lower local distortion favors a long-range transfer of kinetic energy by atomic interactions, in which case the cascade process produces more chains of interstitials squeezed in special crystal orientations, that is, crowdions. Through snapshots of defect spatial evolutions during the cascade processes, the present study also revealed that the majority of crowdions in the form of long chains vanished at the end of the simulation, which might have resulted from the reverse of displacement of those interstitial atoms. The transient formation of the crowdions could partially account the remarkable spikes in defect counts in Figure 4, especially with a lower mass fraction of Mo.

Moreover, the present study also examined the average atomic fractions of Mo among atoms with a displacement of more than 1 Å under different Mo contents, as shown in Figure 7b. Ignoring the case of pure U, Mo fractions in displaced atoms have always been observed to be lower than that in the total simulation bulk, which could also serve as a corroboration of the negative role of Mo in kinetic energy transfer. It could then be inferred that the introduction of Mo hinders the swelling of the displacement spike and thus restricts the spatial distribution of defects, which results in a higher probability of defect annihilation and a partial decrease of the residual defect population.

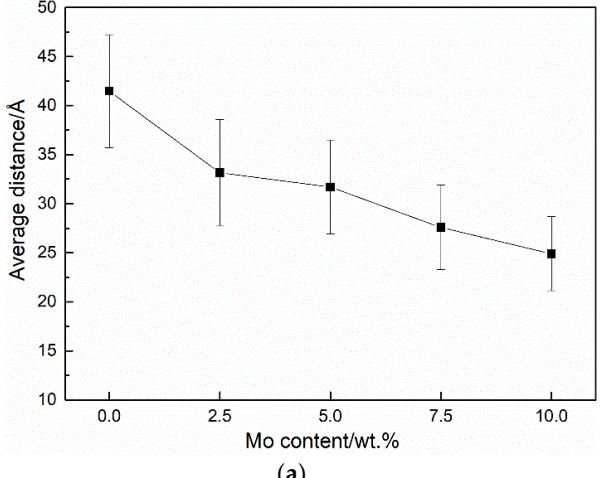
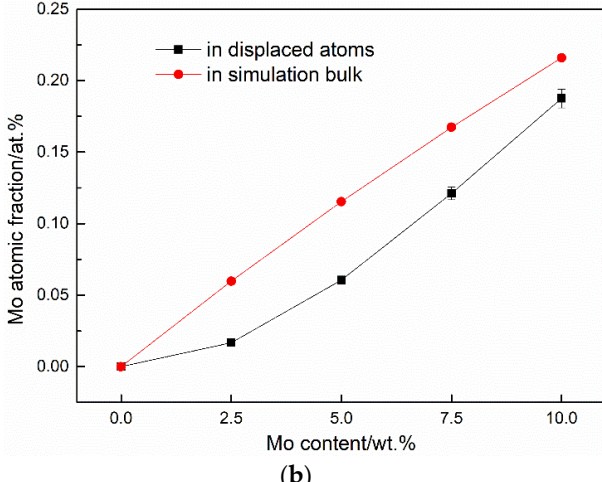

**Figure 7.** (**a**) Average distances between residual defects and the initial position of PKA as a function of Mo content and (**b**) Average atomic fractions of Mo among atoms with displacement more than 1 Å as a function of Mo content. The corresponding Mo atomic fractions in the total simulation bulk are also shown for comparison.

## 5. Conclusions

In the present research, an ADP was developed to capture atomic behaviors in U-Mo systems. Macroscopic properties including lattice constants, cohesive energies, and elastic constants were reproduced moderately well. Meanwhile, reasonable corrections were also implemented in the intermediate range of the potential, with a good agreement with DFT results. Moreover, the newly constructed potential was applied to simulations of primary radiation damage in U-Mo alloys and an uplift of residual defect population was observed when Mo content decreases to 5 wt.%. This indicates an increase of defect evolution and serves as a corroboration of the critical role of Mo depletion near GBs in the onset of IIR, and of accelerated swelling behavior in nuclear fuels.

**Author Contributions:** Conceptualization, W.L., J.L. (Jiahao Li), J.L. (Jianbo Liu), B.L. and W.O.; methodology, W.L. and J.L. (Jiahao Li); software, W.O.; validation, W.O.; formal analysis, W.O.; investigation, W.O.; resources, B.L.; writing—original draft preparation, W.O.; writing—review and editing, J.L. (Jianbo Liu). All authors have read and agreed to the published version of the manuscript.

**Funding:** This research was funded by the National Key Research and Development Program of China, grant number No. 2017YFB0702401, and the National Natural Science Foundation of China, grant number No. 51631005.

**Acknowledgments:** The authors acknowledge Yi Wang and Jingcheng Chen for providing inspiration for dealing with technical problems.

**Conflicts of Interest:** The authors declare no conflict of interest. The funders had no role in the design of the study; in the collection, analyses, or interpretation of data; in the writing of the manuscript, or in the decision to publish the results.

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
