# Peer review of "Atomic Simulations of U-Mo under Irradiation: A New Angular Dependent Potential"

_metals, doi:10.3390/met11071018_

Round 1

Reviewer 1 Report

This is an interesting paper, the newly developed force field could prove useful to better understand the behavior in nuclear fuels. The authors could better highlight originality of their work compared to previous model.

The authors introduced a new force field (Angular Dependent Potential) for Uranium-Molybdenum alloy. This new potential leads to reasonable lattice parameters but more importantly, is was used to simulate the evolution of defects in U-Mo alloy which was then related to the evolution of their mechanical properties during irradiation-induced recrystallization.

Major comment:

Table 1 : the difference with experimental lattice constant are much worst for U using this new model compared with ADP from ref15. Can the authors explained the difference between their model and the previous one from ref15 (and ref16) and how is it an improvement ?

Minor comment:

p6, line 200 : « The text continues here. »

This sentence must be remove.

In Figure 4: what is the unit of the cluster size?

Reviewer 2 Report

This work is devoted to a very useful area - modeling the behavior of the structure of materials under the influence of radiation. Experiments of this kind are very expensive, dangerous, and require specialized equipment. Therefore, the development of modeling methods is very important, which makes it possible to abandon a large number of experiments. Therefore, this kind of work should be welcomed.

There are some comments on the design of the paper.
The annotation should not include a reference to the publication ([16] on line 15).
Lines 213-219 and 222 are meaningless. Literary references must be included in the text of the tables.
In addition, in my opinion, additional materials (Table A1 and Figure A1) should be introduced into the main text. They do not take up much space and it makes no sense to move them outside the text of the paper.
